# Possible favorable lifestyle changes owing to the coronavirus disease 2019 (COVID-19) pandemic among middle-aged Japanese women: An ancillary survey of the TRF-Japan study using the original "Taberhythm" smartphone app

**Koichiro Azuma** [1,2]ⓞ*, **Tetsuya Nojiri**[3]ⓞ, **Motoko Kawashima**[4], **Akiyoshi Hanai**[3], **Masahiko Ayaki**[4], **Kazuo Tsubota**[4,5], **on behalf of the TRF-Japan Study Group**¶

**1** Department of Medicine, Nerima General Hospital and Institute of Healthcare Quality Improvement, Public Interest Incorporated Foundation Tokyo Healthcare Foundation, Tokyo, Japan, **2** Institute for Integrated Sports Medicine, Keio University School of Medicine, Tokyo, Japan, **3** Oishi Kenko Incorporated, Tokyo, Japan, **4** Department of Ophthalmology, Keio University School of Medicine, Tokyo, Japan, **5** Tsubota Laboratory Inc., Tokyo, Japan

ⓞ These authors contributed equally to this work.
¶ Membership of the TRF-Japan Study Group is listed in the Acknowledgments.
* azumakx@keio.jp

## Abstract

Coronavirus disease 2019 (COVID-19) has had a global effect on people's lifestyles. Many people have become physically inactive and developed irregular eating patterns, which leads to unhealthier lifestyles and aggravation of lifestyle-related diseases; these in turn increase the severity of COVID-19. Prior to the COVID-19 pandemic, we developed a smartphone application called "Taberhythm" to investigate eating patterns, physical activity, and subjective feelings of happiness. We aimed to compare lifestyle data before and during the first phase of the COVID-19 pandemic to objectively assess lifestyle changes during quarantine. A total of 464 smartphone users (346 women, 35 ± 12 years old, body mass index [BMI] 23.4 ± 4.5) participated in Period A (January 7 to April 28, 2019) and 622 smartphone users (533 women, 32 ± 11 years old, BMI 23.3 ± 4.0) participated in Period B (January 6 to April 26, 2020). Compared with Period A, there was a sharp decline in physical activity during Period B (4642 ± 3513 vs. 3814 ± 3529 steps/day, p<0.001), especially during the final 9 weeks in both periods (4907 ± 3908 vs. 3528 ± 3397 steps/day, p<0.001); however, there were large variations in physical activity among participants. We found a surprising trend during Period B toward increased happiness among women aged 30–50 years, the group most affected by stay-at-home policies that led to working from home and school closure. Moreover, daily eating duration declined in this population. Additionally, there was a positive association of happiness with steps per day in Period B (ρ = 0.38, p = 0.02). Despite the many negative effects of the COVID-19 pandemic, subjective feelings of happiness among

**Data Availability Statement:** All relevant data are within the manuscript and its Supporting Information files.

**Funding:** This study was supported by a grant from Keio University School of Medicine Departmental Teaching and Research Allowance (KT). Oishi Kenko Incorporated developed the smartphone app and covered all expenses of developing the app, as well as the costs of processing and analyzing the data. The CEO and an employee of Oishi Kenko Incorporated had central roles in the conceptualization of the study (TN) and in the collection and analysis of data (AH). TN has received directors' compensation from this company. AH has received a salary as a full-time employee of Oishi Kenko Incorporated.

**Competing interests:** The CEO and an employee of Oishi Kenko Incorporated had central roles in the conceptualization of the study (TN) and in the collection and analysis of data (AH). TN has received directors' compensation from this company. AH has received a salary as a full-time employee of Oishi Kenko Incorporated. This does not alter our adherence to PLoS ONE policies on sharing data and materials. There are no patents, products in development or marketed products associated with this research to declare. The app was developed as a research tool; in the future, the app will be used to support users in maintaining healthier daily eating rhythms.

middle-aged Japanese women tended to increase, which indicates that some favorable lifestyle changes that could be adopted during quarantine in the ongoing COVID-19 pandemic.

# Introduction

The outbreak of coronavirus disease 2019 (COVID-19), which occurred in China in December 2019, was declared a global pandemic by the World Health Organization in March 2020 and has rapidly spread all over the world, including Japan. The first COVID-19 case in Japan was identified on January 15, 2020, and private companies subsequently started to introduce a remote working policy prior to official stay-at-home recommendations issued on February 20. Schools were closed on March 2 and on April 7, a state of emergency was declared, and stay-at-home requests extended.

COVID-19 is not necessarily lethal for all infected individuals; however, lifestyle-related diseases such as obesity, diabetes, and hypertension appear to increase its severity and mortality risk [1]. A cohort study in the United Kingdom showed that a healthier lifestyle that included regular eating patterns with healthier food choices and increased physical activity, which together form a key strategy to prevent lifestyle-related diseases, reduces the risk of COVID-19 hospital admission [2].

However, the COVID-19 pandemic has changed lifestyles dramatically, with many people working from home and having little contact with people other than family members. These changes have possibly led to less physical activity, altered rhythms of daily life, and unhealthier lifestyles. Many adults who are not leaving home to go to work and are spending more time at home may have greatly diminished levels of daily physical activity or time spent outdoors. Additionally, they may be snacking more and experiencing more circadian rhythm disorders. One international online survey identified unhealthier food consumption and meal patterns, as well as decreased physical activity and increased sedentary time, during quarantine [3].

COVID-19 may also adversely affect psychological health [4]. An Italian online survey by Casagrande et al. identified poor sleep quality, high anxiety, and high distress in more than one-third of 2291 respondents [5]. However, quarantine has also been associated with positive effects such as enhanced communication and increased feelings of closeness to family members [6]. Another Italian survey during April 2020 showed a slight increase in physical activity and healthier food choices in some people [7], which indicates a wide variety of lifestyle changes during quarantine.

Although the reasons for the different effects of quarantine on lifestyle or psychological health are unclear, age and sex differences have been reported. An Indian online survey showed an improvement in healthy meal consumption patterns and a restriction of unhealthy food items, especially in younger people (aged <30 years). These findings are similar to the above-cited Italian survey, which showed higher adherence to the Mediterranean diet among participants aged 18–30 years compared with younger and older participants [7], although the Indian study identified a reduction in physical activity coupled with an increase in daily screen time, especially among men [8]. Conversely, a nationwide Brazilian online survey showed that unhealthy food consumption occurred mostly among young adults (aged 18–29 years), with increased sweet snack consumption among women during the pandemic compared with before the pandemic [9]. A cross-sectional survey of a representative adult sample in the Netherlands showed that the lifestyle (including eating behavior) of older participants (≥65 years) was less likely to be changed by lockdown compared with a working age group [10], which

suggests that younger adults, especially those aged 18–29 years, are more prone to both healthy and unhealthy dietary lifestyle changes during quarantine.

Casagrande et al. found that being a women or younger than 30 years was associated with increased generalized anxiety [5] and women or those younger than 50 years were more susceptible to psychological distress at the beginning of the COVID-19 pandemic [11]. An international prospective study using the sleep–wake patterns questionnaire found that approximately one-sixth of healthy volunteers showed a completely desynchronized pattern during the stay-at-home period; most of the desynchronized participants were older than 50 years and men [12].

However, most studies [3, 5–11] are cross-sectional COVID-19-specific surveys, which may contain substantial selection or recall bias; therefore, more objective measures are needed.

The increasing penetration of smartphones and wearable devices means that these technologies are attractive methods of continuously and remotely monitoring people's health and lifestyle [13]. Using an online health platform to collect lifestyle data from smartphones and wearable activity trackers, Sun et al. observed a general later shift in sleep–wake patterns, longer homestays, and fewer daily steps during quarantine compared with before the pandemic [14]. Moreover, they described differences in behavioral changes among European countries, possibly owing to the different focus in COVID-19 interventions [14]. Data from wrist-worn wearable sensors in a longitudinal population health study in Singapore showed robust changes in rest–activity rhythm, namely, delayed bedtime and a large drop in physical activity [15].

The "Taberhythm" smartphone app was developed by Oishi Kenko Inc. prior to the occurrence of the COVID-19 pandemic. The app was designed to investigate eating patterns, particularly the association between time-restricted eating and body weight and subjective feelings of happiness, with data on dry eye and other conditions. We used the Taberhythm app to collect data and assess lifestyle changes during quarantine by comparing periods before and during the first phase of the COVID-19 pandemic. These data are useful and less biased than those of previous studies, as the original intention of the app was not to assess effects of the pandemic among smartphone users and the data were acquired in a more timely fashion than data from recall-based questionnaires.

In the current study, we sought to identify whether there were lifestyle changes, such as in meal timing and physical activity, during the first phase of the COVID-19 pandemic, by comparing data collected from January to April 2019 and from January to April 2020. As age and sex differences in the effects of the COVID-19 pandemic have been reported [5, 7–12], we hypothesized that behavioral changes during the first phase of the COVID-19 pandemic in Japan might also differ by age and sex. We especially focused on women and men aged 30–50 years, who are the population most likely to be affected by lifestyle changes such as working at home and closure of their children's schools.

## Methods

### Participants

This retrospective observational study was performed as an ancillary study of the TRF-Japan study, which aims to investigate healthier eating patterns among Japanese people, especially focusing on the effect of time-restricted eating on body weight and eye health [16, 17]. The inclusion criteria were iPhone users aged 20 years or older and the exclusion criterion was being non-resident in Japan. As participants were recruited via a website, the study mainly included those familiar with smartphones and their apps.

Women and men aged 20 years or older were recruited via a website. A total of 464 smartphone users participated who had at least one record on the Taberhythm app between January 7 and April 28, 2019 (Period A). A total of 622 smartphone users had at least one record between January 6 and April 26 (Period B). Very few participants (n = 8) overlapped between Period A and Period B. Participants were asked to record all foods and calorie-containing beverages that they consumed, along with their wake-up times, bedtimes, subjective feelings of happiness, and eye symptoms. Because this was a non-invasive web-based observational study, informed consent was not obtained from study participants. Instead, we stated on the first screen of the smartphone app that this was a research app and all data obtained would be used for the study. We also clearly stated that participation was totally voluntary and could be withdrawn at any time, and that participants' anonymity would be preserved. Only after participants pressed the "Agree" button at the bottom of the screen could they proceed to the next screen of the app. We also provided contact information about the study and an inquiry form on the app. This opt-out study was approved by the institutional review board of Keio University School of Medicine (no. 20170162).

## Taberhythm smartphone application

The Taberhythm smartphone iOS app, written in Japanese, was developed by Oishi Kenko Inc. and is available from the Apple Store for adults living in Japan. Once participants had electronically agreed to participate in the study, they could fully access the app. All participant data were transferred to a web server and analyzed for this study (S1 Fig).

**Definition of wake-up times and bedtimes.** Wake-up times and bedtimes were recorded by participants manually every day, in minutes. For bedtime, records before 5 am were regarded as records for the previous day; a record at 4 am on August 5 was analyzed as a record at 28 pm on August 4. Time was expressed as minutes from midnight; for example, 600 minutes is 10 am and 1200 minutes is 8 pm.

**Definition of mealtimes.** Mealtimes were recorded separately by manually selecting a meal category from five categories: breakfast, lunch, dinner, snack, and drink (except non-calorie-containing beverages). The times for the above meals were recorded when a photo of the meal was taken. There was also an option to record meals by manually selecting the time, in minutes, without a meal photo.

All data for each record were transferred to a web server, and feedback notifications were sent a maximum of five times. The feedback timing could be set individually, to avoid forgetting to record a meal.

**Definition of daily eating duration.** Daily eating duration was defined as the duration, in hours, between breakfast (or the first caloric intake after 5 am and before breakfast) and the last caloric intake (including snacks) before 5 am of the following day. If a record was missing, we could not detect whether the participant had skipped breakfast; therefore, the daily eating duration was not calculated if there was no breakfast record.

**Measuring steps per day.** As an objective assessment of physical activity, we used steps per day, recorded with participants' smartphones. The data were acquired using the iOS basic application Health Care. There was no option to record this measure manually.

**Subjective feelings of happiness.** Subjective feelings of happiness were evaluated with the question "How much happiness did you feel today?" Participants recorded their happiness levels using a visual analog scale every night before going to bed.

## Statistical analyses

Raw data are shown as weekly box plots (Figs 1–3). As there were large differences in the amount of raw data for each participant, weekly averages for each participant were calculated,

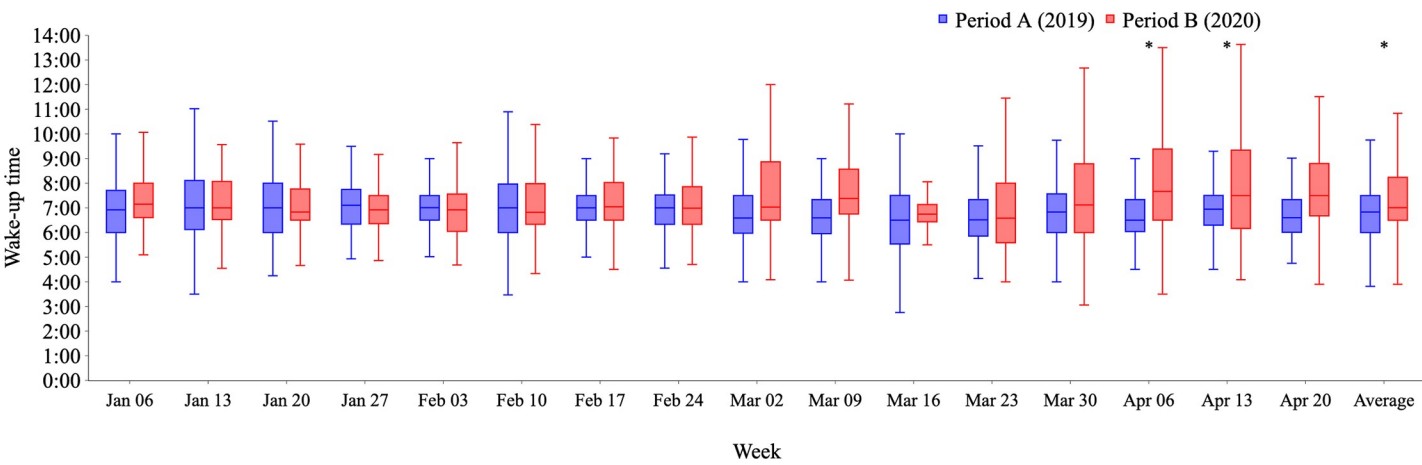

**Fig 1. Weekly changes in wake-up times from Period A (2019) to Period B (2020).** Distributions of all logged data for wake-up times are shown weekly from January 7 to April 28, 2019 (Period A; blue bars) and from January 6 to April 26, 2020 (Period B; red bars). Wake-up time was significantly delayed in weeks 14 and 15 of Period B compared with Period A (both p<0.003), and as the average of 16 weeks, it was also delayed in Period B (p<0.05). *p<0.05/16 ≤ 0.003, using Mann–Whitney U test followed by Bonferroni test.

which greatly reduced data quantity. Therefore, data for each participant are expressed as the averaged data from the 16 weekly averages (Table 1 and Fig 4).

Because we wished to focus on lifestyle changes owing to teleworking and school closures, we hypothesized that women and men aged 30–50 years were most affected by the COVID-19 stay-at-home policies. Therefore, we analyzed these participants' data separately.

We used the Mann–Whitney U test or one-way analysis of variance, as appropriate, to assess group differences. Associations between variables were analyzed using the Spearman test using IBM SPSS Statistics version 21.0 (IBM Corp., Armonk, NY, USA). The Bonferroni test was applied to the 16-weekly group differences to avoid type 1 error from multiple comparisons.

## Results

### Clinical characteristics

As shown in Table 1, participants in Period B were a few years younger than those in Period A (32 ± 11 vs. 35 ± 12 years old, p<0.001).

In both women and men, height and body weight were nearly the same between participants in Period A and B.

### Wake-up times and bedtimes

Irrespective of multiple counts for the same participants, over time, the raw data showed a tendency toward later and larger changes in wake-up time during Period B compared with Period A (Fig 1). There was no apparent trend regarding changes in bedtime during Period A or B (S2 Fig).

When analyzed by participant, wake-up time was later (7:06 am ± 79 min vs. 7:28 am ± 94 min, p = 0.001) during Period B compared with Period A, especially during the final 9 weeks in both periods (7:01 am ± 70 min vs. 7:34 am ± 101 min, p<0.001). There was no sex difference in wake-up time in either period. In women, wake-up time was negatively associated with older age during Period B (ρ = −0.32, p<0.001) whereas no association was observed in Period A. Therefore, in women younger than 30 years, wake-up time was significantly delayed for

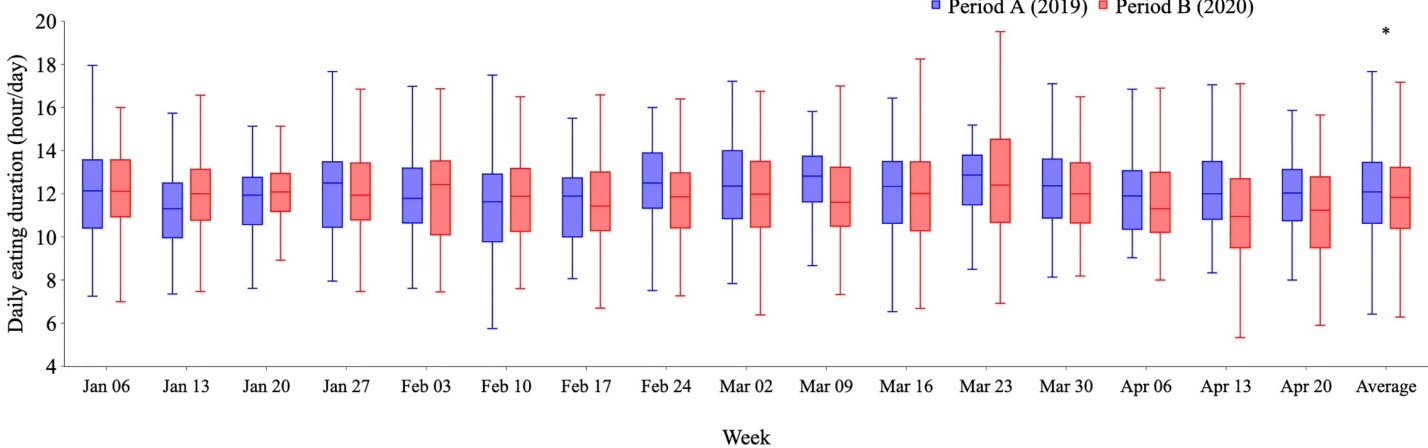

**Fig 2. Weekly changes in mealtimes and daily eating duration from Period A (2019) to Period B (2020).** Distributions of all logged data for breakfast time (a), dinner time (b), and daily eating duration (c) are shown weekly from January 7 to April 28, 2019, (Period A; blue bars) and from January 6 to April 26, 2020 (Period B; red bars). Dinner time (b) was earlier during Period B compared with Period A (p<0.05). Eating duration (c) was shorter in Period B than in Period A (p<0.05).

about 50 minutes during Period B compared with Period A (7:09 am ± 80 min vs. 7:56 am ± 97 min, p<0.001). In men, wake-up time was negatively associated with older age in both Periods A and B (ρ = −0.48, p = 0.001, and ρ = −0.46, p = 0.003, respectively).

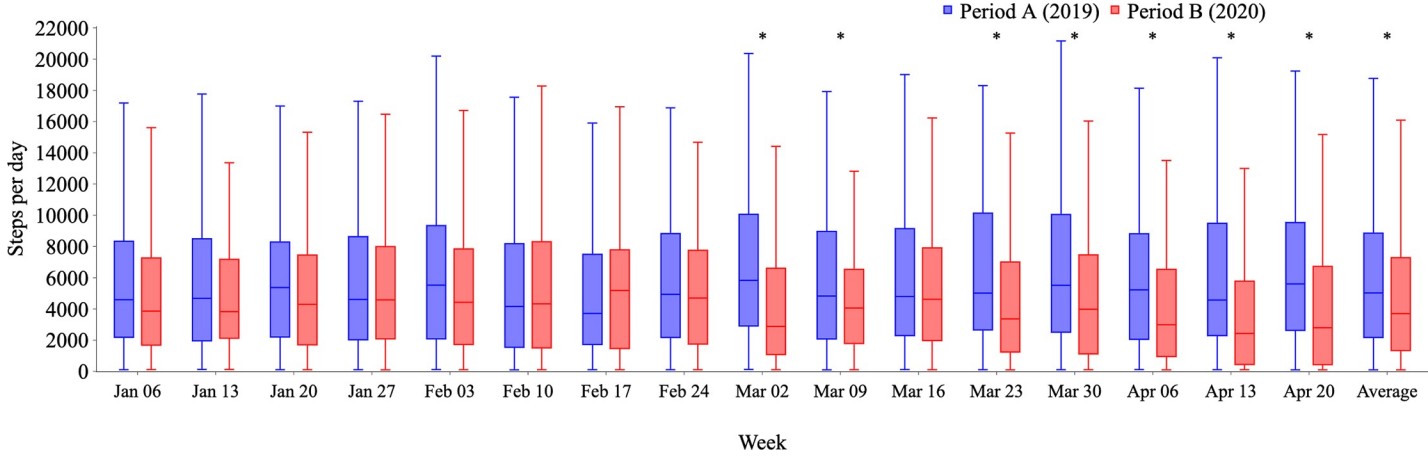

**Fig 3. Weekly changes in physical activity from Period A (2019) to Period B (2020).** Distributions of all logged data for steps per day are shown weekly from January 7 to April 28, 2019, (Period A; blue bars) and from January 6 to April 26, 2020, (Period B; red bars). Physical activity expressed as steps per day was lower in Period B than in Period A, especially in the last 9 weeks of each period.

In both women and men, bedtime was not associated with age and was unchanged from Period A to Period B (0:01 am ± 91 min vs. 11:54 pm ± 81 min).

## Mealtimes

S3 Fig shows all the raw data for mealtimes. Similar to the changes in wake-up times, breakfast timing seemed to gradually become later and more variable in Period B compared with Period A (Fig 2A). Dinner times were approximately 20 minutes earlier in Period B than in Period A (Fig 2B, 7:49 pm ± 102 min for Period A vs. 7:32 pm ± 102 min for Period B, p = 0.003). The daily eating duration was slightly shorter in Period B (Fig 2C, 12.1 ± 2.2 h for Period A vs. 11.9 ± 2.3 h for Period B, p = 0.04).

An analysis by participant showed no difference in mealtimes and daily eating duration between Period A and Period B. Although the number of men in our sample was small, differences in mealtimes by sex were observed, especially among participants aged 30–50 years (Table 2). In women, the timing of lunch was delayed by approximately 30 minutes from Period A to Period B (0:37 pm ± 152 min vs. 1:04 pm ± 113 min, p = 0.04). In women aged 30–50 years, dinner was eaten approximately 30 minutes earlier in Period B than in Period A (7:32 pm ± 89 min vs. 6:58 pm ± 125 min, p = 0.05). In men aged 30–50 years, dinner tended to be eaten approximately 95 minutes later in Period B than in Period A (6:36 pm ± 175 min vs. 8:11

**Table 1. Clinical characteristics of participants.**

| | Period A | | Period B | |
|---|---|---|---|---|
| | **Women, n = 346** | **Men, n = 118** | **Women, n = 533** | **Men, n = 89** |
| Age | 33 ± 11 | 39 ± 14 | 31 ± 10* | 36 ± 12 |
| Height (cm) | 159 ± 5 | 171 ± 6 | 159 ± 6 | 171 ± 7 |
| Body weight (kg) | 57.9 ± 12.8 | 71.7 ± 13.6 | 57.6 ± 10.8 | 71.6 ± 12.3 |
| BMI | 22.9 ± 4.8 | 24.6 ± 4.3 | 22.9 ± 4.2 | 24.5 ± 3.8 |

*p = 0.01 vs. Period A using one-way analysis of variance.

BMI, body mass index.

Period A: January 7 to April 28, 2019; Period B: January 6 to April 26, 2020.

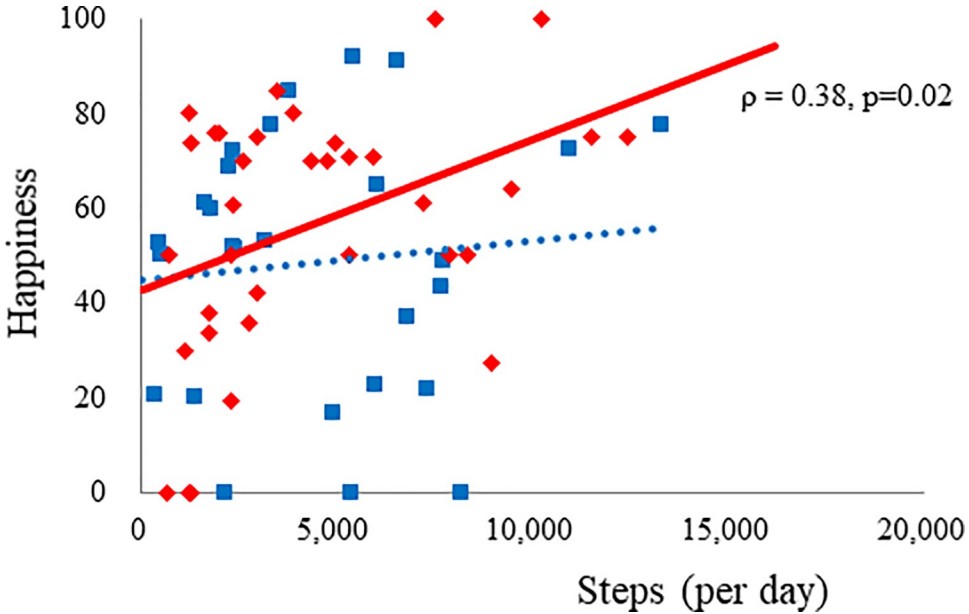

**Fig 4. Associations between physical activity and subjective feelings of happiness among women aged 30–50 years.**
Blue rectangles represent each participant's data for Period A (2019) and red diamonds for Period B (2020). There was a significant positive association between happiness and steps per day in Period B (ρ = 0.38, p = 0.02) but no association in Period A.

pm ± 91 min, p = 0.09). Eating duration was significantly shorter in women aged 30–50 years in Period B compared with Period A (11.1 ± 1.7 h vs. 11.8 ± 1.5 h, p = 0.02).

## Decline in physical activity

As shown in Fig 3, physical activity expressed as steps per day was lower in Period B than in Period A, especially in the last 9 weeks of both periods.

An analysis by participant showed that the number of steps per day decreased by approximately 20% in Period B compared with Period A (4642 ± 3513 vs. 3814 ± 3529, p<0.001), and during the final 9 weeks in both periods, it dropped by approximately 30% (4907 ± 3908 vs. 3528 ± 3397, p<0.001). As shown in Table 3, among women, there was a marginal decline in steps per day in Period B (4198 ± 3206 vs. 3704 ± 3543, p = 0.06), but this was only significant during the final 9 weeks in both periods (4245 ± 3371 vs. 3506 ± 3510, p = 0.04). A decrease in physical activity was less obvious in women aged 30–50 years. In men, there was a significant decline in steps per day in Period B compared with Period A (5827 ± 4008 vs. 4292 ± 3445, p = 0.005), and a further decline during the final 9 weeks (after February 20) in both periods (6309 ± 4572 vs. 3621 ± 2904 p<0.001). There was no significant association between age and steps per day.

## Subjective feelings of happiness

In women aged 30–50 years, happiness tended to be higher during Period B than during Period A (p = 0.11, Table 4). In fact, happiness was significantly higher among women aged 30–50 years than among women aged >50 years or <30 and >50 years during Period B (60 ± 27 vs. 34 ± 35, p = 0.003, 60 ± 27 vs. 49 ± 31, p = 0.02). Happiness was also associated with maintenance of physical activity; that is, there was a significant positive association with steps per day in Period B (ρ = 0.38, p = 0.02, Fig 4) and in the last 9 weeks of Period B (ρ =

**Table 2. Changes in mealtimes and daily eating duration from Period A (2019) to Period B (2020).**

| | Period A | | Period B | |
|---|---|---|---|---|
| | **Women** | **Men** | **Women** | **Men** |
| Breakfast (min) | 515 ± 190 | 491 ± 186 | 526 ± 139 | 551 ± 225 |
| <30, >50 years | (n = 123) | (n = 29) | (n = 206) | (n = 24) |
| Breakfast (min) | 539 ± 179 | 546 ± 238 | 520 ± 161 | 446 ± 240 |
| 30–50 years | (n = 91) | (n = 25) | (n = 143) | (n = 25) |
| Lunch (min) | 755 ± 158 | 783 ± 110 | 785 ± 112 | 786 ± 60 |
| <30, >50 years | (n = 100) | (n = 25) | (n = 191) | (n = 25) |
| Lunch (min) | 761 ± 145 | 791 ± 247 | 783 ± 115 | 724 ± 180 |
| 30–50 years | (n = 74) | (n = 20) | (n = 120) | (n = 24) |
| Dinner (min) | 1159 ± 148 | 1217 ± 60 | 1172 ± 90 | 1156 ± 153 |
| <30, >50 years | (n = 81) | (n = 22) | (n = 144) | (n = 21) |
| Dinner (min) | 1172 ± 89 | 1116 ± 175 | 1138 ± 125[a] | 1211 ± 91[b] |
| 30–50 years | (n = 63) | (n = 13) | (n = 86) | (n = 18) |
| Eating duration (h) | 11.9 ± 2.1 | 13.2 ± 1.8 (n = 19) | 11.7 ± 2.0 (n = 98) | 11.5 ± 3.1 (n = 15) |
| <30, >50 years | (n = 62) | | | |
| Eating duration (h) | 11.8 ± 1.5 (n = 49) | 12.3 ± 1.2 (n = 6) | 11.1 ± 1.7* (n = 65) | 11.7 ± 1.9 (n = 12) |
| 30–50 years | | | | |

*p<0.05

[a] p = 0.05

[b] p = 0.09 vs. 2019 (Period A) using Mann–Whitney U test.

0.53, p = 0.02). In men, happiness increased with age (ρ = 0.42, p = 0.02) in Period A; however, this association disappeared during Period B.

## Discussion

In the present lifestyle log study using a smartphone app, the most striking lifestyle change observed among participants after the start of the COVID-19 pandemic was a decline in

**Table 3. Changes in physical activity from Period A (2019) to Period B (2020).**

| | Period A | | Period B | |
|---|---|---|---|---|
| | **Women** | **Men** | **Women** | **Men** |
| Steps per day | 4292 ± 3197 | 5750 ± 3494** | 3653 ± 3486[a] | 4411 ± 3536[†] |
| <30, >50 years | (n = 152) | (n = 56) | (n = 257) | (n = 50) |
| Steps per day | 4065 ± 3229 | 5933 ± 4665* | 3794 ± 3656 | 4153 ± 3372[†] |
| 30–50 years | (n = 107) | (n = 41) | (n = 142) | (n = 43) |
| After Feb 20 | 4330 ± 3475 | 6285 ± 3979** | 3475 ± 3491[†] | 3845 ± 2708[††] |
| <30, >50 years | (n = 98) | (n = 44) | (n = 159) | (n = 29) |
| After Feb 20 | 4109 ± 3220 | 6343 ± 5373* | 3566 ± 3569 | 3390 ± 3127[†] |
| 30–50 years | (n = 61) | (n = 31) | (n = 80) | (n = 28) |

[a] p = 0.07

[†] p<0.05

[††] p<0.001 vs. Period A (2019)

*p<0.05

**p<0.01 vs. women in the same age group and period, using Mann–Whitney U test.

**Table 4. Changes in happiness from Period A (2019) to Period B (2020).**

| | Period A | | Period B | |
|---|---|---|---|---|
| | Women | Men | Women | Men |
| Happiness | 48 ± 31 | 46 ± 30 | 49 ± 31 | 49 ± 29 |
| <30, >50 years | (n = 51) | (n = 21) | (n = 87) | (n = 12) |
| Happiness | 50 ± 30 | 46 ± 35 | 60 ± 27 | 49 ± 28 |
| 30–50 years | (n = 37) | (n = 10) | (n = 57) | (n = 14) |

physical activity, especially in men. We observed several sex and age differences in lifestyle changes. Despite less favorable changes during quarantine, namely, lower physical activity levels, we saw a trend toward increased happiness among women aged 30–50 years along with a 30-minute earlier dinner time and less time spent eating.

Using an online survey, Maugeri et al. showed a decrease in physical activity by approximately 20% and 40% in women and men, respectively, during the COVID-19 pandemic [18]. We have observed a comparable decrease in physical activity, from February 20, when the stay-at-home recommendation was introduced in Japan. Ong et al. longitudinally collected sleep/activity tracker data from 1824 office workers in Singapore beginning before the outbreak and showed a more robust decrease in physical activity, partly because the baseline physical activity level was higher in the study [15]. Another possible explanation is that the strictness of the lockdown between countries may also be a related factor, as the stay-at-home recommendation on February 20 and the subsequent state of emergency in Japan was a "mild" type of lockdown that was not enforceable and non-punitive. In fact, Sun et al. identified differences in behavioral changes during the COVID-19 pandemic between European countries; they used an online health platform to collect data about participants' sleep, physical activity, location, and phone and social app use duration from smartphone sensors and wearable activity trackers [14]. They found that behavioral changes such as homestay duration, steps per day, and physical distancing during the COVID-19 lockdown were less clear in Denmark than in other European countries, possibly because Denmark implemented stricter restrictions on workplaces and public transport but less strict restrictions on staying at home and public events [14].

The effect on mental health of the COVID-19 pandemic and the subsequent quarantine has received much attention and many cross-sectional surveys have been conducted [5, 6, 8, 11, 19, 20]. In particular, women seem more prone to psychological distress during the COVID-19 pandemic [5, 11, 20, 21]. Casagrande et al. showed that being a woman and young age were associated with increased psychological distress at the beginning of the COVID-19 pandemic [5, 11]. In the current study, subjective feelings of happiness did not necessarily decline in women; in fact, they tended to increase during the first phase of the pandemic in women aged 30–50 years, partly because the COVID-19 pandemic was less severe and its mortality rate was much lower in Japan [5].

There are several possible reasons for the favorable changes observed in this study, although we cannot draw firm conclusions. For example, in households with children, school closure may have resulted in parents taking greater care in preparing their children's meals and helping them with schoolwork (despite having to manage their own remote work tasks). Staying at home definitely results in more close contact with children and spouses, which may have led to an increase in positive feelings that outweighed any negative effects of the quarantine. An online survey conducted in China in January and February 2020 among 263 participants with mean ages similar to those in the current study showed some favorable changes, including increased support from friends and family members and greater feelings of closeness to family

members and others, despite other mildly stressful effects [6]. Such changes may partly explain our finding that happiness increased in some of our participants. Another possibility is that women aged 30–50 years are more accustomed to lifestyle changes, such as marriage and childbirth, and can adjust to new lifestyles more rapidly than other groups.

Malkawi et al. showed that mothers' mental health in Jordan was negatively affected by lower income, lower education, unemployment, and residence [22], which indicates that social background may also be important. An Indian online questionnaire-based survey showed that sleep onset–wake-up times and mealtimes were substantially later during quarantine, and this was more pronounced in younger women [23]. This is in contrast with our finding that women aged 30–50 years tended to eat dinner earlier, and suggests that cultural factors such as sex role or position may be related to behavioral changes [23].

An Australian survey of 1491 slightly older adults showed lower physical activity among 49% of participants; negative lifestyle changes, including lower physical activity, were associated with higher levels of depression, anxiety, and stress symptoms [21]. Many other cross-sectional studies have reported the negative effect of reduced physical activity on psychological health [18, 24, 25]. This is consistent with our finding that increased feelings of happiness were associated with higher physical activity levels among women aged 30–50 years during Period B.

A strength of the current study is that the data were collected with no direct intention to assess the effects of the COVID-19 pandemic; therefore, the data are less biased and changes can be detected more clearly than in other COVID-19-specific surveys. Moreover, we found that there was a favorable change in subjective feelings of happiness in at least some populations, which may be helpful in identifying positive lifestyle changes that could be made during the ongoing COVID-19 pandemic.

Unfortunately, data could not be obtained from the same participants in both 2019 (Period A) and 2020 (Period B) and comparisons between the two periods were not made prospectively with the same participants, although the populations in the two periods were similar in age and BMI distribution. Therefore, we were unable to prospectively assess associations among lifestyle changes. Additionally, the survey did not include any questions to assess specific lifestyle changes during quarantine, such as commuting to work, the presence or absence of children or a spouse, and other factors. Therefore, we hypothesized that men and women aged 30–50 years were most likely to be affected by stay-at-home policies; however, we do not know exactly who experienced the greatest lifestyle changes, and stratifications by age and sex were insufficient to characterize the population with respect to specific lifestyle changes during COVID-19 quarantine. We must acknowledge the limitation of selection bias, as people severely adversely affected by the COVID-19 pandemic were unlikely to have had the time or inclination to participate in our study. It is reasonable to assume that participants were from less-affected populations and experienced more favorable changes, such as in happiness. However, this bias may be smaller than in other COVID-19-specific surveys. Because our inclusion criteria were individuals with any data records during either period, another limitation is that the data were incomplete, and missing data made detailed analyses difficult. Finally, subjective feelings of happiness were evaluated using only one indicator, a visual analog scale; validated and standardized questionnaires such as the Subjective Happiness Scale [26] should be used in future studies. In future interventions, basic information on changes such as teleworking, participation in outdoor physical activities, and the presence or absence of children or a spouse should be collected.

Despite these limitations, it is very important to note that at least some people reported feeling happier despite the current difficult situation. This may offer clues about how lifestyles can be improved in the future, and how people can adapt to COVID-19-related changes in society.

## Conclusion

Despite the many negative effects of the COVID-19 pandemic on individual lifestyles, subjective feelings of happiness have not necessarily decreased in all affected individuals. In this study, which was not a COVID-19-specific survey, we found that some populations, such as middle-aged female Japanese participants, reported experiencing more favorable subjective feelings using our app. The present findings may be helpful in suggesting favorable lifestyle changes that could be adopted during periods of quarantine in the ongoing COVID-19 pandemic.

## Supporting information

**S1 Fig. Diagram of the smartphone-based system developed to monitor human eating patterns.**
(TIF)

**S2 Fig. Weekly changes in bedtimes from Period A (2019) to Period B (2020).** Distributions of all logged data for bedtimes are shown weekly from January 7 to April 28, 2019 (Period A; blue bars) and from January 6 to April 26, 2020 (Period B; red bars). There was no significant change in bedtimes between Period A and Period B.
(TIF)

**S3 Fig. Mealtime scatterplots.** All the raw data for mealtimes are shown weekly as scatterplots from January 7 to April 28, 2019 (Period A) and from January 6 to April 26, 2020 (Period B). Five meal categories are shown as different color plots; blue: breakfast; red: lunch; orange: dinner; purple: snacks; green: drinks (calorie-containing).
(TIF)

**S1 File. Data for weekly averages of wake-up times, bedtimes, mealtimes, and steps per day.**
(PDF)

**S2 File. Data for weekly averages of daily eating duration.**
(PDF)

## Acknowledgments

The TRF-Japan Study Group was established by staff at the Department of Ophthalmology, Keio University School of Medicine, Oishi Kenko Incorporated, and Tsubota Laboratory Incorporated.

Members of the group: Kazuo Tsubota (lead author, tsubota@z3.keio.jp), Department of Ophthalmology, Keio University School of Medicine, Japan; Tsubota Laboratory Incorporation, Tokyo, Japan. Motoko Kawashima and Masahiko Ayaki, Department of Ophthalmology, Keio University School of Medicine, Tokyo, Japan. Koichiro Azuma, Institute for Integrated Sports Medicine, Keio University School of Medicine, Tokyo, Japan. Tetsuya Nojiri, Akiyoshi Hanai, Shota Narisawa, Mitsuo Ishikawa, and Daisuke Matsuoka, Oishi Kenko Incorporated, Tokyo, Japan.

We greatly appreciate the help of all staff in the study group for their efforts in developing the smartphone app and organizing the study. We thank Analisa Avila, ELS, and Diane Williams, PhD, of Edanz Group (https://en-author-services.edanz.com/ac) for editing a draft of this manuscript.

## Author Contributions

**Conceptualization:** Tetsuya Nojiri.

**Data curation:** Akiyoshi Hanai.

**Formal analysis:** Koichiro Azuma, Akiyoshi Hanai.

**Funding acquisition:** Kazuo Tsubota.

**Investigation:** Kazuo Tsubota.

**Methodology:** Motoko Kawashima, Akiyoshi Hanai, Masahiko Ayaki.

**Project administration:** Motoko Kawashima.

**Resources:** Tetsuya Nojiri.

**Software:** Akiyoshi Hanai.

**Supervision:** Masahiko Ayaki, Kazuo Tsubota.

**Validation:** Akiyoshi Hanai.

**Visualization:** Akiyoshi Hanai.

**Writing – original draft:** Koichiro Azuma.

**Writing – review & editing:** Motoko Kawashima, Masahiko Ayaki, Kazuo Tsubota.

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
