## [Decision Letter · Decision Letter 0]

1 Dec 2020

PONE-D-20-29557

Possible favorable lifestyle changes owing to the coronavirus disease 2019 (COVID-19) pandemic among middle-aged Japanese women: an ancillary survey of the TRF-Japan study, using the original “Taberhythm” smartphone app

PLOS ONE

Dear Dr. Azuma,

Thank you for submitting your manuscript to PLOS ONE. After careful consideration, we feel that it has merit but does not fully meet PLOS ONE’s publication criteria as it currently stands. Therefore, we invite you to submit a revised version of the manuscript that addresses the points raised during the review process.

We look forward to receiving your revised manuscript.

Kind regards,

John William Apolzan, PhD

Academic Editor

PLOS ONE

Journal Requirements:

"Oishi kenko Incorporated developed the smartphone app and covered all expenses of developing the app, as well as costs for processing and analyzing the data; the funder will use the app for promotion of their site."

We note that one or more of the authors are employed by a commercial company: Oishi kenko Incorporated.

2.1. Please provide an amended Funding Statement declaring this commercial affiliation, as well as a statement regarding the Role of Funders in your study. If the funding organization did not play a role in the study design, data collection and analysis, decision to publish, or preparation of the manuscript and only provided financial support in the form of authors' salaries and/or research materials, please review your statements relating to the author contributions, and ensure you have specifically and accurately indicated the role(s) that these authors had in your study. You can update author roles in the Author Contributions section of the online submission form.

2.2. Please also provide an updated Competing Interests Statement declaring this commercial affiliation along with any other relevant declarations relating to employment, consultancy, patents, products in development, or marketed products, etc.  

3. One of the noted authors is a group or consortium [TRF-Japan Study Group]. In addition to naming the author group, please list the individual authors and affiliations within this group in the acknowledgments section of your manuscript. Please also indicate clearly a lead author for this group along with a contact email address.

Reviewers' comments:

Reviewer's Responses to Questions

**Comments to the Author**

1. Is the manuscript technically sound, and do the data support the conclusions?

Reviewer #1: Partly

Reviewer #2: Yes

2. Has the statistical analysis been performed appropriately and rigorously? 

Reviewer #1: Yes

Reviewer #2: Yes

3. Have the authors made all data underlying the findings in their manuscript fully available?

Reviewer #1: Yes

Reviewer #2: Yes

4. Is the manuscript presented in an intelligible fashion and written in standard English?

Reviewer #1: No

Reviewer #2: Yes

5. Review Comments to the Author

Reviewer #1: Review PONE-D-20-29557

Possible favorable lifestyle changes owing to the coronavirus disease 2019 (COVID-19) pandemic among middle-aged Japanese women: an ancillary survey of the TRFJapan study, using the original “Taberhythm” smartphone app

The objective of the study was to compare lifestyle data between periods before and after the start of the COVID-19 pandemic, to objectively assess lifestyle changes during quarantine. A total of 464 smartphone users (346 women, 35±12 years old, body mass index [BMI] 23.4±4.5) participated in period A (7 January to 28 April 2019) and 622 smartphone users (533 women, 32±11 years old, BMI 23.3±4.0) participated during period B (6 January to 26 April 2020).

General comment

It is important to assess changes in lifestyles during the COVID-19 pandemics but the present study is somewhat confusing and difficult to read due to an excessive use of figure and poor phrasing. Overall, the data presented must be greatly simplified for clarity. Figures 1, 3a, 3b, 3c and 4 should include two separate line (A and B) and lesser values on the X-axis. Figure 2a and 2b should be removed (correlation coefficient is enough). Figure S3 is unreadable. Also, there are only 9 references cited in the whole manuscript, indicating that authors failed to properly review the others studies on the COVID-19 consequences, the use of app to collect data, the influence of sociodemographic variables on lifestyles, the feeling of happiness, and so on…As a result, the introduction, the research question and the study objectives are quite poorly developed. Consequently, the discussion is too short and quite mundane.

Specific comments:

The introduction lacks of a sound theoretical framework. Many studies have been published since March 2020 about the effects of COVID-19 pandemics and confinement on lifestyles and mental health in the general population. Also studies investigating the added value of health-related app to collect data on lifestyles should be mentioned to justify the interest of the present study, and the research question.

Please replace “individuals” with “participants” in the text.

Line 86: We applied the Taberhythm app to collect objective data and assess lifestyle changes during quarantine by comparing periods before and after the COVID-19 pandemic” However, only the number of step per day can be considered as “objective measure”. Please reformulate.

Line 95. “Our findings can be helpful in suggesting positive lifestyle habits that can be adopted during quarantine periods in the ongoing COVID-19 pandemic” this sentence anticipate the result. Replace this sentence with clear study objectives, based on properly documented instroduction.

Inclusion and exclusion criteria must be clarified. The characteristics of the main study participants should be mentioned.

The study design must be specified

It is unclear whether or not the same participants were included in waves A et B.

Line 165: “Because we wished to focus on lifestyle changes owing to teleworking and school closures, we hypothesized that women and men age 30–50 years were most affected by the COVID-19 stay-at-home policies. Therefore, we analyzed these individuals separately”. The evidence (data) underlying this hypothesis must be mentioned in the introduction.

Also, authors performed descriptive analyses separated by sex. This must be justified in the introduction. And waves A and B should be in columns, while covariates (sex…) in row, for clarity.

The results are confusing and difficult to read, as mention in general comments.

As mentioned before, the discussion should be more developed based on presented results, theoretical inputs, and insights from previous studies.

Reviewer #2: This paper investigates how COVID-19 has affected people’s lifestyle, in particular, eating, walking, and the sense of happiness. This is done by analysing and comparing data collected in 2019 and 2020 through a smartphone app Taberhythm. While this topic is of general interests and considerable social implications, the authors may need to clarify or improve upon the points listed below.

1) It is not clear to readers why the authors chose the starting date to be early January, given that Japanese governments put forward recommendations in mid Feb 2020. The inclusion of the period where COVID-19 was much less of concern may complicate the analysis and results.

2) The authors need to specify the reason why they believe the decline in eating duration was a result of earlier eating

3) the authors wrote, “younger women (age 20–25 years) participated in the study slightly more than during January to April 2020 (Period B) than during January to April 2019 (Period A); therefore, participants’ mean age in Period B was a few years younger than in the previous year (32±11 vs. 35±12 years old, p<0.05).”

I am not sure if it can be concluded that the overall younger age was due to a higher number of people in very specific age range (20 – 25 yrs). Plus, there are more women enrolled in 2020 anyway.

4) The authors need to be made aware of a work done in Europe covering very similar behaviours such as bedtime and walking, entitled ‘Using Smartphones and Wearable Devices to Monitor Behavioral Changes During COVID-19’. It would be interesting to compare these results in different continents and cultures.

5) the authors need to increase the resolution of all figures. It is hard to see details at the moment. It is also necessary to reconsider the presentation of Figure 1,3,4. The boxes are significantly overlapping, making it very difficult for readers to read.

6) There are occasional grammatical errors the authors need to correct

6. PLOS authors have the option to publish the peer review history of their article (what does this mean?). If published, this will include your full peer review and any attached files.

Reviewer #1: No

Reviewer #2: No

---

## [Author Response · Author response to Decision Letter 0]

3 Feb 2021

Responses to the comments of the Academic Editor

[Comment 1] Please ensure that your manuscript meets PLOS ONE's style requirements, including those for file naming. The PLOS ONE style templates can be found at https://journals.plos.org/plosone/s/file?id=wjVg/PLOSOne_formatting_sample_main_body.pdf and https://journals.plos.org/plosone/s/file?id=ba62/PLOSOne_formatting_sample_title_authors_affiliations.pdf

Response: We have ensured that our manuscript meets PLOS ONE’s style requirements, including those for file naming. We have modified the description of author roles according to the CRediT Taxonomy as follows: 

Conceptualization: T. Nojiri, K. Azuma. Data curation: A. Hanai. Formal analysis: K. Azuma, A. Hanai. Funding acquisition: K. Tsubota. Investigation: T. Nojiri, M. Kawashima, M. Ayaki. Methodology: T. Nojiri. Project administration: T. Nojiri, M. Kawashima, K. Tsubota. Resources: T. Nojiri, A. Hanai. Software: A. Hanai. Supervision: K. Tsubota. Validation: K. Azuma, M. Kawashima, M. Ayaki. Visualization: K. Azuma, T. Nojiri. Writing: original draft preparation: K. Azuma. Writing: review and editing: M. Ayaki, M. Kawashiwa, K. Tsubota.

[Comment 2-1] Thank you for stating the following in the Competing Interests section: "Oishi kenko Incorporated developed the smartphone app and covered all expenses of developing the app, as well as costs for processing and analyzing the data; the funder will use the app for promotion of their site." We note that one or more of the authors are employed by a commercial company: Oishi kenko Incorporated.

Please also include the following statement within your amended Funding Statement.　“The funder provided support in the form of salaries for authors [insert relevant initials], but did not have any additional role in the study design, data collection and analysis, decision to publish, or preparation of the manuscript. The specific roles of these authors are articulated in the ‘author contributions’ section.”

Response: Oishi Kenko Incorporated developed the smartphone app and covered all expenses of developing the app, as well as the costs of processing and analyzing the data. The CEO and an employee of Oishi Kenko Incorporated had central roles in the conceptualization of the study (T. Nojiri) and in the collection and analysis of data (A. Hanai). T. Nojiri has received directors’ compensation from this company. A. Hanai has received a salary as a full-time employee of Oishi Kenko Incorporated.

[Comment 2.2] Please also provide an updated Competing Interests Statement declaring this commercial affiliation along with any other relevant declarations relating to employment, consultancy, patents, products in development, or marketed products, etc. 

Response: The current study was designed purely for academic interest and was completely unrelated to employment, consultancy, patents, products in development, or marketed products of the company (Oishi Kenko Incorported). The app was developed as a research tool; in the future, the app will be used to support users in maintaining healthier daily eating rhythms. This does not alter our adherence to PLOS ONE policies on sharing data and materials.

[Comment 3] One of the noted authors is a group or consortium [TRF-Japan Study Group]. In addition to naming the author group, please list the individual authors and affiliations within this group in the acknowledgments section of your manuscript. Please also indicate clearly a lead author for this group along with a contact email address.

Response: We have listed the individual authors and affiliations within this group and indicated a lead author (K. Tsubota) in the acknowledgments section. 

Responses to the comments of Reviewer #1

[General comment] It is important to assess changes in lifestyles during the COVID-19 pandemics but the present study is somewhat confusing and difficult to read due to an excessive use of figure and poor phrasing. Overall, the data presented must be greatly simplified for clarity. Figures 1, 3a, 3b, 3c and 4 should include two separate line (A and B) and lesser values on the X-axis. Figure 2a and 2b should be removed (correlation coefficient is enough). Figure S3 is unreadable. Also, there are only 9 references cited in the whole manuscript, indicating that authors failed to properly review the others studies on the COVID-19 consequences, the use of app to collect data, the influence of sociodemographic variables on lifestyles, the feeling of happiness, and so on…As a result, the introduction, the research question and the study objectives are quite poorly developed. Consequently, the discussion is too short and quite mundane. 

Response: We have corrected Figures 1, 3a, 3b, 3c, and 4 according to your advice. We have deleted Figure 2. All the figures (including Figure S3) have been clarified. We have reviewed the references on COVID-19 consequences, including behavioral changes, psychological health, and the use of apps to collect data, and have included an additional 17 references. In the introduction section, the rationale of focusing on age and sex differences in changes during quarantine has been expanded. In the discussion section, the effect of sociodemographic variables on lifestyle, and different approaches to the COVID-19 pandemic among countries, have been described.

[Specific Comment 1] The introduction lacks of a sound theoretical framework. Many studies have been published since March 2020 about the effects of COVID-19 pandemics and confinement on lifestyles and mental health in the general population. Also studies investigating the added value of health-related app to collect data on lifestyles should be mentioned to justify the interest of the present study, and the research question.

Response: We have reviewed previous studies on the effect of the COVID-19 pandemic on lifestyle and mental health and have added some references to the third and fourth paragraphs of the introduction section. We have added a paragraph discussing wearable sensor technologies to collect objective lifestyle data to the fifth paragraph of the introduction section. We have also added a paragraph on the need to discuss age and sex differences (fourth paragraph of the introduction section).

[Specific Comment 2] Please replace “individuals” with “participants” in the text.

Response: We have replaced “individuals” with “participants” throughout the manuscript.

[Specific Comment 3] Line 86: We applied the Taberhythm app to collect objective data and assess lifestyle changes during quarantine by comparing periods before and after the COVID-19 pandemic” However, only the number of step per day can be considered as “objective measure”. Please reformulate.

Response: We have reformulated the introduction by adding a paragraph on longitudinal studies using sensor technologies (lines 115–124 in the revised manuscript with tracked changes) and compared them with cross-sectional online surveys (lines 76–111 in the revised manuscript with tracked changes). We have deleted the word “objective” from the sentence on line 86 (line 129 in the revised manuscript with tracked changes).

[Specific Comment 4] Line 95. “Our findings can be helpful in suggesting positive lifestyle habits that can be adopted during quarantine periods in the ongoing COVID-19 pandemic” this sentence anticipate the result. Replace this sentence with clear study objectives, based on properly documented introduction.

Response: We have replaced this sentence with “We hypothesized that behavioral changes during the first phase of the COVID-19 pandemic in Japan might also differ by age and sex” (lines 139–140 in the revised manuscript with tracked changes) and have added a new paragraph reviewing previous reports showing age and sex differences in the effects of the COVID-19 pandemic (lines 89–111 in the revised manuscript with tracked changes).

[Specific Comment 5] Inclusion and exclusion criteria must be clarified. The characteristics of the main study participants should be mentioned.

Response: We have clarified the inclusion and exclusion criteria as follows: “The inclusion criteria were iPhone users aged 20 years or older and the exclusion criterion was being non-resident in Japan” (lines 151–152 in the revised manuscript with tracked changes). The characteristics of the main study participants were described as “As participants were recruited via a website, the study mainly included those familiar with smartphones and their apps” (lines 152–154 in the revised manuscript with tracked changes).

[Specific Comment 6] The study design must be specified

Response: We have added the terms “retrospective observational” (line 148 in the revised manuscript with tracked changes).

[Specific Comment 7] It is unclear whether or not the same participants were included in waves A et B.

Response: Only eight participants overlapped between Period A and Period B. We have mentioned this limitation in the discussion section (lines 440–444 in the revised manuscript with tracked changes). We have clearly stated this as follows: “Very few participants (n = 8) overlapped between Period A and Period B” (lines 158–159 in the revised manuscript with tracked changes).

[Specific Comment 8] Line 165: “Because we wished to focus on lifestyle changes owing to teleworking and school closures, we hypothesized that women and men age 30–50 years were most affected by the COVID-19 stay-at-home policies. Therefore, we analyzed these individuals separately”. The evidence (data) underlying this hypothesis must be mentioned in the introduction.

Response: We have added a paragraph about age and sex differences in behavioral changes during quarantine (lines 89–111 in the revised manuscript with tracked changes) and suggested that younger adults aged 18–30 years were more prone to make both healthy and unhealthy lifestyle changes, whereas older adults were more susceptible to desynchronized sleep–wake cycles during quarantine.

[Specific Comment 9] Also, authors performed descriptive analyses separated by sex. This must be justified in the introduction. And waves A and B should be in columns, while covariates (sex…) in row, for clarity.

Response: We have added a paragraph about age and sex differences in behavioral changes during quarantine and suggested that men tended to become more sedentary (lines 95–96 in the revised manuscript with tracked changes) and that women were more susceptible to greater psychological distress (lines 105–107 in the revised manuscript with tracked changes). 

We have reformulated the tables so that Period A and B are in columns and covariates such as age are in rows.

[Specific Comment 10] The results are confusing and difficult to read, as mention in general comments.

Response: We have deleted Figure 2. The weekly raw data (Figures 1–3) and averaged data by participant (Tables 2–4, Figure 4) have been presented separately for clarity. Data for the final 8 weeks (the 2nd half of the period) were deleted. Instead, as the stay-at-home recommendation started on February 20, data for the final 9 weeks of the period (from February 20) are shown for wake-up times (lines 257–259 in the revised manuscript with tracked changes) and physical activity (lines 326-339 in the revised manuscript with tracked changes).

[Specific Comment 11] As mentioned before, the discussion should be more developed based on presented results, theoretical inputs, and insights from previous studies.

Response: We have reviewed more references and on the basis of this previous research, we have discussed the decline in physical activity (lines 376–394 in the revised manuscript with tracked changes) and increased happiness among women aged 30–50 years (lines 397–400 and lines 419-426 in the revised manuscript with tracked changes). 

Responses to the comments of Reviewer #2

This paper investigates how COVID-19 has affected people’s lifestyle, in particular, eating, walking, and the sense of happiness. This is done by analysing and comparing data collected in 2019 and 2020 through a smartphone app Taberhythm. While this topic is of general interests and considerable social implications, the authors may need to clarify or improve upon the points listed below.

 [Comment 1] It is not clear to readers why the authors chose the starting date to be early January, given that Japanese governments put forward recommendations in mid Feb 2020. The inclusion of the period where COVID-19 was much less of concern may complicate the analysis and results.

Response: From the present perspective, the study should have started after mid-February 2020. However, when we started the study, we felt there had been substantial lifestyle changes from the beginning of 2020. Therefore, we added the following sentence: “The first COVID-19 case in Japan was identified on January 15, 2020, and private companies subsequently started to introduce a remote working policy prior to official stay-at-home recommendations issued on February 20” to the introduction section (lines 59–61 in the revised manuscript with tracked changes). As this study began at the beginning of the COVID-19 pandemic, we replaced the text “after the COVID-19 pandemic” with “during the first phase of the COVID-19 pandemic” in the abstract section, the introduction section, and the discussion section.

As you kindly pointed out, there were large drops in physical activity after February 20. We have added analyses for the final 9 weeks of the period (from February 20). This enabled us to delete the analyses for the 2nd half (the final 8 weeks) of the period, which greatly simplified the results (Table 3 and lines 326-339 in the revised manuscript with tracked changes). We also observed greater delay in wake-up times from February 20. We have also added data from February 20 for wake-up times (lines 257–259 in the revised manuscript with tracked changes).

[Comment 2] The authors need to specify the reason why they believe the decline in eating duration was a result of earlier eating

Response: Eating duration is mostly defined according to breakfast time and dinner time. Therefore, if breakfast time had not changed, earlier dinner times would shorten the eating duration. However, because snacks may have been consumed, and not all participants had both breakfast time and dinner time data, we could not fully explain the change in eating duration. Therefore, we have deleted the text “mainly owing to eating dinner 30 minutes earlier than usual” from the abstract section (lines 48-49 in the revised manuscript with tracked changes) and “as a result” from the the results section (line 286, line 294, and line 305 in the revised manuscript with tracked changes).

[Comment 3] the authors wrote, “younger women (age 20–25 years) participated in the study slightly more than during January to April 2020 (Period B) than during January to April 2019 (Period A); therefore, participants’ mean age in Period B was a few years younger than in the previous year (32±11 vs. 35±12 years old, p<0.05).”

I am not sure if it can be concluded that the overall younger age was due to a higher number of people in very specific age range (20 – 25 yrs). Plus, there are more women enrolled in 2020 anyway.

Response: We agree with your point and have deleted the text “younger women (age 20–25 years) participated in the study slightly more than during January to April 2020 (Period B) than during January to April 2019 (Period A)” from the results section (lines 231–233 in the revised manuscript with tracked changes).

[Comment 4] The authors need to be made aware of a work done in Europe covering very similar behaviours such as bedtime and walking, entitled ‘Using Smartphones and Wearable Devices to Monitor Behavioral Changes During COVID-19’. It would be interesting to compare these results in different continents and cultures.

Response: We have added citations to this study to the introduction section and the discussion section (lines 117–122 and lines 386–394 in the revised manuscript with tracked changes). In the introduction section, we have mentioned the importance of wearable sensor-based technologies to remotely collect lifestyle data, instead of using online surveys. In the discussion section, we have discussed sociodemographic differences, such as strictness of the lockdown, which may have resulted in differences in behavioral changes among countries. 

[Comment 5] the authors need to increase the resolution of all figures. It is hard to see details at the moment. It is also necessary to reconsider the presentation of Figure 1,3,4. The boxes are significantly overlapping, making it very difficult for readers to read.

Response: We have increased the resolution of all figures and separated the boxes in Figures 1, 3, and 4 (now 1–3). 

[Comment 6] There are occasional grammatical errors the authors need to correct

Response: We have corrected the grammatical errors. We have shown only major changes to content and expression, and minor grammatical changes have not been tracked to avoid making the manuscript difficult to read.

---

## [Decision Letter · Decision Letter 1]

9 Mar 2021

Possible favorable lifestyle changes owing to the coronavirus disease 2019 (COVID-19) pandemic among middle-aged Japanese women: An ancillary survey of the TRF-Japan study using the original “Taberhythm” smartphone app

PONE-D-20-29557R1

Dear Dr. Azuma,

We’re pleased to inform you that your manuscript has been judged scientifically suitable for publication and will be formally accepted for publication once it meets all outstanding technical requirements.

Kind regards,

John William Apolzan, PhD

Academic Editor

PLOS ONE

Additional Editor Comments (optional):

If the authors wish to further clarify the eating duration term, they can but overall seems sufficient.

Reviewers' comments:

Reviewer's Responses to Questions

**Comments to the Author**

1. If the authors have adequately addressed your comments raised in a previous round of review and you feel that this manuscript is now acceptable for publication, you may indicate that here to bypass the “Comments to the Author” section, enter your conflict of interest statement in the “Confidential to Editor” section, and submit your "Accept" recommendation.

Reviewer #1: All comments have been addressed

Reviewer #2: (No Response)

2. Is the manuscript technically sound, and do the data support the conclusions?

Reviewer #1: Yes

Reviewer #2: Yes

3. Has the statistical analysis been performed appropriately and rigorously? 

Reviewer #1: Yes

Reviewer #2: Yes

4. Have the authors made all data underlying the findings in their manuscript fully available?

Reviewer #1: Yes

Reviewer #2: Yes

5. Is the manuscript presented in an intelligible fashion and written in standard English?

Reviewer #1: Yes

Reviewer #2: No

6. Review Comments to the Author

Reviewer #1: (No Response)

Reviewer #2: The definition of eating duration is somewhat confusing. One would expect the total amount of time spent on eating. I am not sure how the authors intend to make sense of this parameters. The authors may consider elaborating it more, and preferably using another term with less ambiguity.

7. PLOS authors have the option to publish the peer review history of their article (what does this mean?). If published, this will include your full peer review and any attached files.

Reviewer #1: **Yes: **Aymery Constant

Reviewer #2: No

---

## [Editor Report · Acceptance letter]

12 Mar 2021

PONE-D-20-29557R1 

Possible favorable lifestyle changes owing to the coronavirus disease 2019 (COVID-19) pandemic among middle-aged Japanese women: An ancillary survey of the TRF-Japan study using the original “Taberhythm” smartphone app 

Dear Dr. Azuma:

I'm pleased to inform you that your manuscript has been deemed suitable for publication in PLOS ONE. Congratulations! Your manuscript is now with our production department. 

Kind regards, 

on behalf of

Dr. John William Apolzan 

Academic Editor

PLOS ONE